# Clustering Accelerometer Activity Patterns from the UK Biobank Cohort

**DOI:** 10.3390/s21248220

**Published:** 2021-12-09

**Authors:** Stephen Clark, Nik Lomax, Michelle Morris, Francesca Pontin, Mark Birkin

**Affiliations:** 1Leeds Institute for Data Analytics and School of Geography, University of Leeds, Leeds LS2 9JT, UK; n.m.lomax@leeds.ac.uk (N.L.); f.l.pontin@leeds.ac.uk (F.P.); m.h.birkin@leeds.ac.uk (M.B.); 2Leeds Institute for Data Analytics and School of Medicine, University of Leeds, Leeds LS2 9JT, UK; m.morris@leeds.ac.uk

**Keywords:** accelerometer, wearables, personal activity, clustering, profiling

## Abstract

Many researchers are beginning to adopt the use of wrist-worn accelerometers to objectively measure personal activity levels. Data from these devices are often used to summarise such activity in terms of averages, variances, exceedances, and patterns within a profile. In this study, we report the development of a clustering utilising the whole activity profile. This was achieved using the robust clustering technique of k-medoids applied to an extensive data set of over 90,000 activity profiles, collected as part of the UK Biobank study. We identified nine distinct activity profiles in these data, which captured both the pattern of activity throughout a week and the intensity of the activity: “Active 9 to 5”, “Active”, “Morning Movers”, “Get up and Active”, “Live for the Weekend”, “Moderates”, “Leisurely 9 to 5”, “Sedate” and “Inactive”. These patterns are differentiated by sociodemographic, socioeconomic, and health and circadian rhythm data collected by UK Biobank. The utility of these findings are that they sit alongside existing summary measures of physical activity to provide a way to typify distinct activity patterns that may help to explain other health and morbidity outcomes, e.g., BMI or COVID-19. This research will be returned to the UK Biobank for other researchers to use.

## 1. Introduction

Being physically active is known to promote healthy outcomes, ranging from prevention of non-communicable diseases, such as type 2 diabetes, osteoporosis, dementia, and certain cancer prevention [1], to mental well-being [2] and even reduced risk of severe COVID-19 outcomes [3]. However, 23% of adults globally are not sufficiently active; the World Health Organisation launched the Global Action plan to tackle physical inactivity in 2018. This plan calls for systems to establish the sociocultural and environmental determinants of physical inactivity, so that they can subsequently be tackled [4]. In the developed world, where manual employment is less prevalent, physical inactivity rates are higher, with 34% of men and 42% of women in the United Kingdom (UK) not active enough for good health [5]. As a result, one in six UK deaths can be attributed to physical inactivity [5], the same as attributed to smoking. This comes not only at a personal cost with respect to health, but also a financial cost of GBP 7.4 billion annually in the UK [5].

Physical activity makes up the energy expenditure side of the energy balance equation. Given the low levels of activity observed around the world, it is not surprising that we also observe high levels of obesity. Improving physical activity levels and reducing obesity prevalence is one mechanism through which health can be improved, but the benefits of physical activity go far beyond weight status [5]. A pertinent example has come to the forefront since the COVID-19 pandemic struck, with higher levels of physical activity being associated with lower severity of COVID-19 symptoms [6]. In addition to population health, there are substantial planetary health benefits associated with being more active, and replacing car journeys with active travel alternatives.

In part, it is challenging to understand physical activity behaviours and their drivers because they are difficult and expensive to measure, with many surveys using subjective self-reported activity diaries. Objective measures of physical activity from accelerometers are valuable as they are less susceptible to the issues of self-report faced by surveys or interview, which rely upon participant recall and, therefore, can be subject to inaccurate recall or purposeful misreporting due to social desirability bias [7]. In addition, they are able to capture extensive and fine-grained temporal nuances in behaviours, which enable us to begin to unpick the drivers of physical activity and understand the balance between activity and inactivity. Nonetheless, due to the associated costs of collecting accelerometry data, study size and representativeness are often limited [8]. The UK Biobank cohort study is one of the first studies to achieve objective measures of physical activity at a scale not normally achieved by surveys [9].

As part of the UK Biobank initiative, a subset of over 103,000 participants wore wrist worn accelerometer devices for 7 days, and data were subsequently processed using the Euclidean Norm Minus One (ENMO) metric and made available to researchers [10]. Data were recorded as activity level over a 5 s time period (measured as milli-gravity units, mg), providing 120,960 data points for each participant.

These data have subsequently been used in combination with the UK Biobank’s extensive data on disease and health outcomes [11]. These included studies into: functional fitness, measured through walking pace [12] or grip strength [13]; cardio vascular disease [14,15] and heart failure [16]; cancer generally [17] and specifically breast cancer [18]; mental health [19] including psychiatric conditions [20,21] and depression and anxiety [22]; mortality generally [23,24], but also concentrating on volume and intensity of activity [25,26]; sleep quality [27,28] and the relationship of sleep to morbidity outcomes [6,29]; and circadian rhythms [30,31]. Outside of health, a range of studies have been reported that correlate these objectively measured physical activity levels with other observations and outcomes such as environmental characteristics [32], travel [33], and activity [34,35].

To utilise these data, many studies attempt to summarise the activity patterns, often using statistical summary measures of the activity level, with the risk that using such summary measures loses some of the richness and value in these data. The average level of activity is the most common measure used [36,37,38]. Another study, concerned with Parkinson’s disease, used the variation in movement to act as proxies for the subject’s gait and lateral movement [39]. Others have used various thresholds based on milli-gravities to identify periods of activity covering a range of levels, from being sedate, through light and moderate, to vigorous activities [16,22,40]. Another approach is to concentrate on just the higher moderate to vigorous physical activity range [41,42,43,44].

The aim of this study is to provide an alternative metric to classify the activity of individuals. We illustrate the use of processed raw accelerometer data to produce a useful and comprehensive measure, and investigate its importance to health. In particular, rather than adding to the existing continuous measures of physical activity we provide a categorical based measure of physical activity. This measure is defined by both the participant’s pattern of activity over the week and the intensity of this activity, capturing both periods of physical activity and sedentary behaviours. We then look at the variation in participant’s characteristics, based on sociodemographic attributes, socioeconomic status, their health, and circadian rhythm. We additionally demonstrate the utility of these patterns to differentiate amongst participants on two outcomes, body mass index (BMI) [45] and COVID-19 testing outcome. This is similar to what others have done when exploring how aggregate and threshold derived measures affects outcomes such as physical fitness, cardiovascular disease etc. (see above). We hypothesise that (1) BMI will be higher in less active activity patterns and (2) positive COVID-19 test outcomes will be higher in those who are least active and reliant on outside care.

## 2. Materials and Methods

The UK Biobank is a prospective study of over 500,000 participants aged between 40 and 69 years, resident in Great Britain and recruited via their general medical practitioner service. Participants were asked to attend one of 22 regional assessment centres during the period 2007 to 2010 to collect baseline data. During this visit, data were collected on various themes, including: sociodemographic, family history, lifestyle behaviours, and health status. In addition, some physical measurements were taken: height, weight, blood pressure, grip strength, spirometry, and fitness tests, along with biomarker data such as blood and urine samples. Participants were asked to consent to linkages to their historic and future electronically held heath data—this latterly including COVID-19 testing and outcome data. A comparison of the make-up of these participants and the general population is reported in Fry, Littlejohns [46].

Subsequently sub-samples of participants were asked back for repeat assessments and also to complete various on-line surveys, for example around diet [47], occupational history [48], and mental health [49]. One important request was for participants to wear wrist worn accelerometers for one week over the period June 2013 to December 2015. The processing of these data is described in Doherty, Jackson [10].

### 2.1. Activity Profile Selection

Our research team was supplied with accelerometer data from 103,687 participants, summarised into 5 s time periods. This measure of physical activity was provided in milli-gravity units (mg), with higher values indicating greater activity, and to put this measure into context, a table of mg values for typical daily activities is provided by Migueles, Cadenas-Sanchez [50], with a 3 min walk at a usual speed, measuring around 130 mg on an accelerometer worn on the dominant wrist. Over a full week, there should have been 120,960 of these 5 s time periods; however, the completeness of these data varied by participants. We grouped the participants into five different groups according to the quality and completeness of these data, as described below:

Complete and perfect. Where participants provide valid readings for all 120,960 of the 5 s time periods during the week of wear. These data were used as reported, with no adjustment.

Complete and imputed. Where participant data required some imputation by UK Biobank, using readings from the same time-of-day on other days [10], thereby providing the 120,960 readings. These data were used as reported, with no adjustment.

Incomplete, but recoverable. These data were provided with 120,959 readings (either actual or UK Biobank imputed), with additionally just one blank reading, typically the final 5 s of the reporting period. These data are used here, but with a recognition that when readings were averaged over 1 h (see below), for one of the hours this is based on 719 readings rather than 720.

Switch to day light savings time. This happens in the UK in late March, as the time zone moves from Greenwich Mean Time (GMT) to British Summer Time (BST), effectively providing blank data from 01:00:00 GMT to 02:00:00 BST. These participants provide 121,680 readings, with at least 720 of these being blank and the final day’s wear extended by 1 h. To use these data, the missing blank hour of data on Sundays from 01:00:00 GMT to 01:59:55 GMT was be imputed by us using the average of readings in this same hour on other days and the extra hour of data at the end of the series removed.

Switch from daylight saving time. This happens in late October, effectively duplicating data during 01:00:00 BST to 01:59:00 BST with 01:00:00 GMT to 01:59:00 GMT data, providing just 120,240 readings. This means that these data have one less hours’ worth of data on the final day, usually 09:00:00 GMT to 09:59:55 GMT, and to recover these data, this final hour is imputed by us using the averages of readings in this same hour on other days.

The data from the participants who did not fall into one of these five categories was not used in this study. Further participant data were not used (even though it was technically useable) because the participant had asked to withdraw from the study, or they were younger than 40 or older than 69 at baseline assessment and, thus, outside the scope of UK Biobank.

In line with other studies, these useable data were further checked for suitability. Participants whose average activity levels were greater than 100 mg were excluded and those with less than 72 h of non-impute wear were removed; both thresholds have been used in other studies involving these data [15,16,51]. Recognising that there is some noise in these data, for later analysis purposes, these 5 s time periods were averaged out to cover 1 h periods (an average over 60 × 12 = 720 5 s time periods).

As can be seen in the Section 3, this process provided us with over 91,000 useable activity profiles. The goal was then to establish whether there are any meaningful clusters and what their overall activity patterns look like.

### 2.2. Clustering Algorithm Choice

For this study, a number of clustering techniques were considered. One consideration was the number of participants present in these data. Techniques that required the computation of a full (but symmetric) distance matrix (of approximately 91,533 by 91,533 in size, ≈4140 million entries) were too computationally demanding for meaningful analysis. Additionally methods that were themselves computationally expensive, in terms of calculating these distances, would also present challenges. These two considerations lead us to looking at the k-means family of clustering algorithms.

These methods only require the computation of similarities (measured as distances) from each participant to a set of k cluster centres (i.e., a sequence of n × k calculations, rather than calculations to populate a triangular n × (n − 1)/2 matrix). Consideration is then needed of how to best calculate these similarities and to what an ‘averaging’ might look like to establish the cluster centres. Some studies have advocated the use of dynamic time warping (DTW) distances as an alternative to Euclidean distances in measuring the similarity between time series trajectories [52,53]. However the calculation of DTW distances is much more time consuming than the Euclidean equivalents [54]. Genolini, Ecochard [55] highlight these issues, and propose the use of a custom distance metric, an extension of the Fréchet distance, to compute these distances and a Fréchet mean to calculate cluster centres. Recognising that these refinements can be computationally demanding, even for small to medium sized data sets, they propose workarounds, such as initial clustering using Euclidean distance to generate a large sample of representative trajectories (e.g., k = 128), which are then themselves clustered using a Fréchet based approach, or by simplifying the trajectories themselves. Both these solutions involve assumptions on the representativeness of the initial clustering solution and compromises on the detail within the trajectories.

Working on an assumption that the Euclidean distance is good enough to capture the similarity between activity profiles, we used a random sample of our data to compare the patterns in the distance matrix calculated using both DTW and Euclidean distances. We require that the pattern in these two matrices should be similar, e.g., two profiles that are similar using the DTW metric should also be similar using the Euclidean metric. If we are satisfied with this assumption, then k-means and its variants become possibilities for analysis. We assess these metrics in Section 3.2.

Acknowledging that the activity profiles can be noisy, even when averaged to hourly profiles, we adopted the k-medoids clustering algorithm [56,57]. The k-medoid algorithm works in a similar manner to k-means except that the cluster centres are chosen to be actual profiles rather than an aggregation of profiles. This approach makes the algorithm less sensitive to the presence of unusual profiles, since it is effectively using the median rather than the mean as its measure of average. It also uses the Euclidean distance rather than the squared Euclidean distance, which again reduces the influence of outlying profiles.

### 2.3. Model Specification and Validation

To fit these models, the K_Medoids function from the R package ClusterR was used [58], running on a high performance computer with 40 cores and 192 Gb of RAM. A measure of dis-similarity is calculated that measures how heterogeneous each cluster is, with the aim to minimise this value. However, by definition, this value will always decrease as the number of clusters, k, increases (as with the scree plot in k-means [59]). An approach to deciding the number of clusters is to look for an elbow in a plot of k versus the dis-similarity measure. This elbow indicates a change in the gradient in the plot, where the reduction from having additional clusters levels off. Another measure to evaluate the clustering performance is that the sizes of each cluster should be relatively similar, with no cluster having too many or too few participants associated with it. The chosen value of k can be further internally validated by examining the resultant activity profiles, to see if they are plausible. External validation can be gained by seeing how these clusters are characterised against other information available on the participant that was not used in the clustering process.

### 2.4. Application of Activity Profiles to Health Outcomes

To illustrate the utility of these activity profiles, two outcomes were examined to see if they vary between the clusters, a participant’s BMI and COVID-19 test result. This is possible since at the baseline visit the participant’s standing height and weight were objectively measured by a nurse, with these measures then used to calculate their BMI. Here, we report the activity patterns associated with each participant’s BMI status.

The UK Biobank recently provided information to researchers on various outcomes related to COVID-19. One rich and contemporary source of information concerns testing, whether a participant had undergone a test for COVID-19, and what the test result was. Here, we have data for tests conducted up to 23 March, 2021, which was after the UK’s second wave of infections. It is also in this period—up to the end of March 2021—that most of the UK Biobank participants, who would be in priority groups by virtue of their age, would have been offered at least one of their two COVID-19 vaccinations. We report the number of participants in each activity pattern that were alive on 23 March, 2021, the percentage of these participants who had a test, the proportion that tested positive, and the percentage of tests that were positive.

## 3. Results

The UK Biobank provided 103,687 acceleration profiles from volunteer participants. Subsequent to wearing the accelerometer, nine participants asked for their data to be removed from UK Biobank. Of those remaining, 346 profiles were from participants who were aged younger than 40 or older than 70 at baseline and therefore should technically not be part of the cohort. This leaves 103,332 activity profiles for potential analysis (99.7% of the original 103,687).

### 3.1. Selection of Profiles

As described above not all these profiles will be amenable to analysis. Looking at those that are available, we find that there are 22,056 profiles that are complete and perfect (21% of the 103,332); 45,574 that are complete and imputed (44%); 21,634 that are incomplete but recoverable (21%); 1141 that occurred during the switch to daylight savings time (1%); and 2015 that occurred during the switch from daylight savings time (2%). This provides 92,420 profiles for potential analysis (89% of the 103,332).

The 16 participants that have an average activity level of 100 mg or more and the 887 who have less than 72 h of valid wear are removed from our sample, leaving 91,533 profiles useable for analysis (88% of the original 103,687 supplied to us by UK Biobank). Figure 1 provides examples of two profiles where the average activity levels are (a) consistently high and (b) contain high spikes, with Appendix A showing the pattern for all 15.

In Table 1 the composition of the full UK biobank sample, the sample of those who wore an accelerometer, and those whose data were useable for clustering are presented to illustrate if there are any bias introduced as a result of this selection process. There are clear differences between the full UK Biobank sample and those who volunteered to wear an accelerometer. Those in the accelerometer sample are more likely to be female, white British, in employment, car users, with higher levels of education and be wealthy. They also have better health, but their circadian rhythms are not that dissimilar. The composition of the sample useable for clustering is broadly similar to those that wore an accelerometer across each of the characteristics. This provides reassurance that the process of identifying and cleaning the 103,332 available activity profiles into a useable sample has not biased the resultant sample when measured on these metrics.

### 3.2. Comparison of Distance Metrics

Here, we present a comparison of the distance (or dis-similarity) calculated using: (a) Euclidean metric or (b) DTW metric. For 40 randomly selected participants, the full distance matrix was calculated and the values from these calculations are illustrated in Figure 2.

An initial inspection shows that whilst the actual distance values are different (the DTW values are generally higher), the general patterns in both matrices are very similar. Pairs of participants that are far apart on one metric (large blue circles) are also far apart on the other, and pairs that are close on one metric (small red circle) are also close on the other. This provides the reassurance that the computationally simpler and quicker Euclidean distance is an acceptable substitute for the more computationally expensive DTW calculation.

### 3.3. K-Medoids Clustering

K-medoid models were fitted to the 91,533 activity profiles for various values of k, ranging from k = 1 to k = 18. The dis-similarity scree plot and the reductions in dis-similarity, as k increases, are shown in Figure 3.

The identification of an elbow in Figure 3a is difficult to discern, however the first difference plot in Figure 3b provides some guidance on a suitable value for k. There are substantial reductions in the dis-similarity moving from k = 1 to k = 4. The reduction from moving from k = 4 to k = 5 is much smaller, and for the next movement, from k = 5 to k = 6 this reduction stays constant. Further drops in this reduction occur moving from k = 6 to k = 7 and from k = 8 to k = 9. Thereafter the reductions are similar and generally level. Recalling that this first difference is akin to a gradient, Figure 3b suggests that after k = 9 the gradient becomes linear, and the reductions in dis-similarity are constant and not a function of k. We see that the last time that a sizeable reduction in dis-similarity occurred was moving from k = 8 to k = 9. This then suggests that there are potentially nine meaningful clusters in our activity patterns.

### 3.4. Nine Cluster Solution Verification

Having arrived at a solution with nine clusters, the plausibility and differentiation of the resultant overall patterns of activity associated with each cluster helps to validate the choice of the number of clusters. For the nine-cluster solution, the average activity patterns within each cluster is shown in Figure 4, with descriptive labels below each chart. Additionally, the medoids and average profiles for all cluster solutions k = 1 to k = 16 are provided in the Appendix A. Here we introduce the nine clusters and provide a short descriptive label for each cluster. These labels are for convenience to aid interpretation and commonly used in clustering work [60,61]. Care has been taken to minimise subjectivity and potential stigma.

The first group has 8313 participants who have a pattern with bursts of activity in the mid-morning and late afternoon during the weekdays, with a plateau in between. They are also active on a weekend, particularly in the morning. These have been termed our “Active 9 to 5” participants because of their high levels of activity and their regular weekday patterns, suggestive of activity on the way to—and on the way home from—work. The second group are our straight-forward “Active”, who have similar levels of activity to the “Active 9 to 5”, maintaining an active day, but the 9 to 5 pattern is not present, missing the late-afternoon spike. Our third group is termed “Morning Movers” because they exhibit a burst of activity in the morning, but this is not maintained and quickly diminishes through the day. The “Get up and Active” group also have a spike of activity in the morning, as seen in all our other patterns so far, but they plateau a level of moderate activity during the day, unlike the “Morning Movers” whose activity levels do not plateau.

The fourth group is those who appear to “Live for the Weekend” and are the first to not show a morning spike, instead maintaining a plateau of activity through each weekday. However, at the weekend their activity levels do spike, particularly on a Sunday morning. Our “Moderates” group is perhaps the most average of the patterns. There are no discernible spikes in the pattern, but the activity levels through the day are maintained and moderate. The “Leisurely 9 to 5” group looks to be similar to our first “Active 9 to 5” group, but with some important differences, firstly the activity levels are much lower for these participants, being nearly half as active, and there does not appear to be a spike of activity at the weekends. The eighth group is our “Sedate” group who have lower levels of activity that quickly diminish through the day. The final group is our “Inactive” group who have a pattern with the lowest levels of activity and no discernible spikes.

These patterns associated with each group appear to be plausible and no group is particular large or small, with the largest group, “Get up and Active” making up 17% of the participants and the smallest, “Inactive”, being 7%.

### 3.5. Nine Cluster Solution Validation

Whilst the previous section demonstrates that the clusters can be differentiated using the activity profiles that were used to drive the clustering, further insight and reassurance can be gained by seeing how participants in each of these activity pattern groups can be characterised against other information. Fortunately, the UK Biobank contains a rich variety of information on each participant that is suitable for this purpose.

Here, we use information from a number of domains (as shown in Table 1) to characterise the participants in each group. Table 2 shows the percentage of participants within each group that fall into each category, with rows highlighted so that the largest value is in red whilst the lowest is in blue (the full tables are provided as Appendix A). So for example, 61.1% of the ‘Active 9 to 5’ group of participants are female, whilst only 45.3% of the ‘Inactive’ group are female.

Table 1 identifies clear differential patterns, summarised below.

Active 9 to 5. These participants are predominately female, younger, and from a White British background. They have high levels of employment, qualification, and income. They are very healthy, with a healthy BMI, and few other reported health issues. Their circadian rhythms suggest that they are mostly able to get up in the morning. They also tended to wear the devices during the spring/summer seasons.

Active. Participants in this group are similar to the “Active 9 to 5” participants, but have a slightly different circadian rhythm; they have difficulty getting up in the morning and definitely do not see themselves as morning people.

Morning Movers. These participants are again largely female, but not so young. They are less likely to be in employment and have lower levels of education, whilst living in deprived neighbourhoods. Their health is generally better than the clustering sample as a whole. They find it very easy to get up in the morning, but they do not necessarily consider themselves morning people.

Get up and Active. This group is similar to those who are “Morning Movers”, but are slightly more likely to be Female and are better educated.

Live for the Weekend. The sex split for this group is not dissimilar to the entire clustering sample, but they are slightly younger and less White British. They are more likely to be in paid employment and have reasonably good levels of education, wealth, and health. Their circadian rhythm is also fairly typical of the clustering sample.

Moderates. These participants are generally in line with the entire clustering sample, but distinctively different when it comes to reporting having problems getting up in the morning and do not generally identify as morning people.

Leisurely 9 to 5. Here, we see a greater proportion of males. This is a relatively young group, with high levels of employment, education, and wealth. Their health is not so good however, with a low percentage having a healthy BMI.

Sedate. Participants in this group are predominately male, older, and White British. They have lower employment and levels of education, but they live in less deprived neighbourhoods. They were more often asked to wear the device in autumn or winter.

Inactive. This group has the largest percentage that is male. Whilst older, they are not that much older than participants in some of the other patterns. They also report low percentages for the socioeconomic characteristics, such as employment, education, and wealth. Their health is poor.

One of the strongest trends in Table 1 is that showing the composition of each group by sex. Whilst females do generally have a higher representation in the clustering sample, they are very much more represented in the more active patterns. Otherwise the groupings accord with prior expectations in that the more active patterns are younger, have better socioeconomic characteristics, are in better health.

### 3.6. Obesity Outcomes

As noted in the Introduction, levels of physical activity can have an important influence on measures of health and wellbeing. One such measure is the BMI. Table 3 shows how the BMI status of participants in each group distributes amongst the activity patterns. Looking at this table, we see that those who are a healthy weight or who are overweight are most likely to have a “Get up and Active” pattern, whilst those who are living with obesity are most likely to be “Sedate”.

To better understand these trends relative to the group sizes, Table 3 also provides the difference in these percentages-so of those participants who have a healthy BMI, 13.1% are in the “Active 9 to 5” group, which is +4.0% more than the equivalent percentage across all obesity outcomes, here 9.1%. Here we now see that whilst 19.9% of those who are living with obesity are in the “Sedate” group, this is only 5.7% more than the percentage across all obesity outcomes, whilst the 15.3% who are “Inactive” are 7.9 percentage points more than the expected 7.4%. Looking at these differences, we see that the healthier patterns are more likely to be over represented with healthy BMI outcomes (positive differences) whilst the less healthy patterns are more likely to be over represented in the outcome of those living with obesity. The differences for those who are overweight are much less pronounced.

### 3.7. COVID-19 Test Outcomes

In Table 4, we report the number of participants in each pattern that were alive on 23 March 2021, the percentage of these participants who had a test, the proportion that tested positive and the percentage of tests that were positive. This table shows that mortality is highest with the least active patterns, which is unsurprising given their age and health profile as shown in Table 2. Even given that we are looking at physical activity behaviour at some time before the pandemic, we still observe a relationship between COVID-19 outcomes and historical physical activity levels. The participants in inactive groups are more likely to be tested than participants with other activity patterns, but their test outcomes are more likely to be negative than for participants with most other activity patterns, with only a small percentage of tests returning positive results. The groups with the highest positive test results are the two 9 to 5 patterns, whose positivity rates are high, at nearly a quarter of tests coming back positive.

## 4. Discussion

In this study, a clustering of the whole activity profile of UK Biobank participants has been carried out to produce nine distinct patterns of activity: “Active 9 to 5”, “Active”, “Morning Movers”, “Get up and Active”, “Live for the Weekend”, “Moderates”, “Leisurely 9 to 5”, “Sedate” and “Inactive”.

When we look at the characteristics of participants in each pattern, we observe an over-representation of females in the more active patterns. In their systematic review of studies with a focus on the 50 and older age range, Notthoff, Reisch [62] found the evidence for differing activity patterns between the sexes to be somewhat inconclusive and varied by type of activity, with van Uffelen, Khan [63] highlighting different sex-specific motivating factors to undertake types of physical activity. Previously however, O’Donoghue, Perchoux [64] found women to have a higher mean vector magnitude, a proxy for total physical activity, than men in all but the age group (aged 45–54). Moreover, recent work by Pontin, Lomax [65] looking at trends in habitual physical activity behaviour identified that, despite females moving more throughout the day, it was at a lower intensity and, thus, did not routinely contribute to meeting physical activity guidelines framed around moderate to vigorous activity.

Otherwise, the other characteristics accord with prior expectations, with the active patterns being younger, wealthier, better educated, having good health status, and with circadian rhythms more attuned to the mornings. For the more inactive patterns, these characteristics reverse.

To assess if the season in which a participant wore a sensor might impact cluster membership, we looked at self-reported health outcomes for each group, disaggregated by the season in which members wore an accelerometer. We found no substantial difference, suggesting that season does not disproportionately drive activity pattern membership, and should be interpreted in the same way as the other variables reported in Table 2. A previously study using a similar k-means clustering method identified that weekly activity behaviour patterns contributed to seasonal physical activity; however, the different weekly behaviour patterns were not mutually exclusive to one pattern of seasonal behaviour [66].

With the BMI outcome, there are clear differences in outcomes by our activity patterns, individuals with a healthy BMI are more likely to be in the more active groups whilst individuals who are obese are more likely to be in the ‘Inactive’ and ‘Sedate’ groups. This is unsurprising given that physical activity is one side of the energy balance equation [67]. Additionally, these active UK biobank participants are more likely to have other good lifestyle behaviours and other characteristics previously related to lower BMI than their “Inactive” and “Sedate” counterparts, which accords with a whole systems mapping approach to understanding BMI [68].

Following on from this understanding of BMI, there is a link to our second outcome, testing and test results for COVID-19 either through obesity [69,70] or through ethnicity and other factors [71]. In their study, Chadeau-Hyam, Bodinier [72] found that having a high BMI increased the chances of being tested and testing positive. These earlier studies mainly used test outcomes when the vast majority of testing took place in a hospital setting, with very little community testing taking place, meaning that most test subjects would already have adverse health conditions. In our extended analysis of COVID-19 test outcomes, covering a period with enhanced community testing, we find that those with the historically least active patterns, and therefore having poorer BMI outcomes, are also more likely to be tested, as others have reported. However, conversely we find that these tests are less likely to be positive for these inactive patterns, counter to one of our hypothesis that the inactive would be more likely to test positive. This outcome could, in reality, be a result of those participants with these patterns following government guidelines in regards to shielding [73], or being cautious in who they interact with, reducing their exposure to COVID-19. In fact, the groups with the highest positivity rate are the “Active 9 to 5” and the “Leisurely 9 to 5” patterns. The period for which we have the COVID-19 testing data (up to March 2021) saw periods between lockdowns where more people returned to work and therefore had greater exposure to possible infections. Their activity pattern is certainly suggestive of activities around commute journeys to a place of work, and such activities may expose people to more contacts in environments that they have little control over [74]. In support of this, Rowlands, Dempsey [75] also found that there was a positive association between total physical activity and higher odds of a non-severe infection, with an odds estimate of 1.10.

Our identification of activity behaviour helps provide a novel tool for the research community to compare observed patterns of physical activity behaviour in different contexts and from a wide range of smartphone and wearable sources [76]. A number of studies have performed similar exercises. In a study by Willetts, Hollowell [51], 132 non-UK Biobank adult volunteers were asked to wear both wrist accelerometers and cameras, and these ‘ground truth’ data were used to train a random forest algorithm to identify periods in the day that the volunteer was undertaking certain classes of activity. These activities were sleep, sit/stand, vehicle, mixed-activity, and bicycling. This model was then applied to the equivalent accelerometer data from UK Biobank to identify periods of such activities within the cohort. Their study was able to predict certain activities within the day, in contrast to the approach here, which is to look at the entire week of activity.

Using the approach of classifying time periods within the week, Lam, Catt [77] created a matrix of bouts of high-level activities, where the percentage split, amount of time and number of bouts undertaken in five activity classes during four time periods in the day, were collated for each participant. These high-level summary measures were then used to classify participants using k-means and hierarchical clustering. They identified just three classes, which, like here, they went on to characterise, with the ultimate aim to predict presence of type-2 diabetes using a variety of supervised machine learning algorithms. They found that their study supports the hypothesis that individuals with diagnosed type-2 diabetes exhibit physical activity patterns that are significantly different from those of the control participants.

In a study of the impact of shift work patterns on life expectancy, Pocuca, Farrell [31] identified 5507 UK Biobank participants who undertook shift work and wore an accelerometer, which yielded 5029 activities patterns amenable to analysis. They performed a clustering using a finite mixture model, which predicted three clusters in this specialist study. The conclusion was that there were protective effects from undertaking physical activity, which could offset the additional risk associated with shift work. Whilst this is a more specialist study than the one we report, it does demonstrate the utility of summary clusters in explaining health related outcomes.

Whilst not using UK Biobank data, Jones, Mirkes [78] used equivalent data to train and validate supervised machine learning algorithms to develop a portable model to discriminate activity clusters. They used k-means to identify 10 clusters in their data that were further grouped using information on time spent undertaking nine types of physical activities into ‘super-clusters’ of sedentary (5 clusters), mixed (1 cluster), slow (2 cluster), brisk (1 cluster) and running (1 cluster). The primary aim of this study was to test the portability of its models to other similar datasets, but UK Biobank was not one of these other datasets. There were similarities with our clusters here, in that they identified diverse sedentary groups. The medoids that have been identified here to define each cluster can be used to cluster other similar accelerometer datasets, by identifying which of the nine medoids each activity is closest to.

Looking beyond personal characteristics, there are issues around the built environment that may affect the form of physical activity patterns that people undertake [75]. Using smartphone data, Althoff, Sosic [79] illustrate that the walkability of the city was associated with patterns of average activity, with inhabitants of walkable cities having increase physical activity during both work and leisure time. Whilst our study has concentrated on the characteristics of the participants that exhibit each of our patterns there are some neighbourhood attributes that could be investigated, such as traffic volume and intensity, and access to greenspace [80].

In this study, we chose to keep each weekday as a separate day, which tends to give greater weight to weekdays in the formation of clusters, but does allow differences between weekdays to be accounted for. This differentiation would be important for participants who did not work a regular Monday to Friday shift pattern or did not undertake similar tasks on a daily basis. Investigating differences by weekdays, and using data on activity around mobile phone cell towers in Rome, Italy, Sevtsuk and Ratti [81] modelled and identified differences in the amount of activity by separate weekdays. In many studies, Fridays are often reported as atypical weekdays. Jiang, Ferreira [82] used extensive survey data from Chicago, finding that Friday activities usually differ from those during the other the weekdays, additionally Zhao, Koutsopoulos [83] used smart card derived travel data from London to report longer duration of out of home activities on Fridays. Future research could explore using a more aggregate profile, consisting of an average week day, plus Saturdays and Sundays. In this case, the weekend activity pattern would assume a greater influence on cluster formation.

The rich accelerometer, lifestyle, sociodemographic, and health outcomes information available within cohorts, such as the UK Biobank, present exciting opportunities to identify holistic lifestyle behaviour patterns, and how they can be associated with positive health outcomes. On a population scale, these can then in turn be used to personalise policy recommendations.

## 5. Conclusions

Our study has shown that it is possible to apply machine learning algorithms to processed accelerometer data to identify distinct temporal and intensity patterns of physical activity, which have a utility in differentiating across a range of outcomes, including: all-cause mortality, weight status, and COVID-19 test outcomes. These data will be deposited back to the UK Biobank study and be made available to other researchers for future use. As much as we have done with COVID-19 and BMI outcomes, these patterns can then be used by others to help understand a range of outcomes.

## Figures and Tables

**Figure 1 sensors-21-08220-f001:**
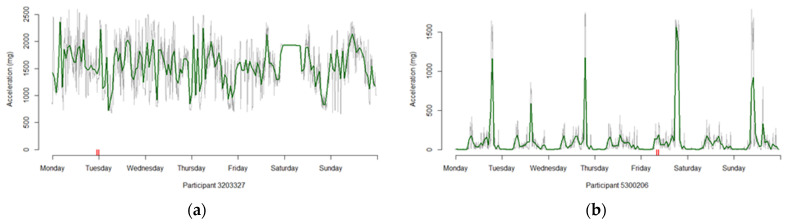
Two profiles with large average activity profiles, with the green line being the hourly profile, the grey line the 5 min profiles and the red notches indicating the presence of imputed values for that time period. (**a**) Participant whose activity levels remain high for the entire 7 days. (**b**) Participant whose activity levels spike to high values on several occasions.

**Figure 2 sensors-21-08220-f002:**
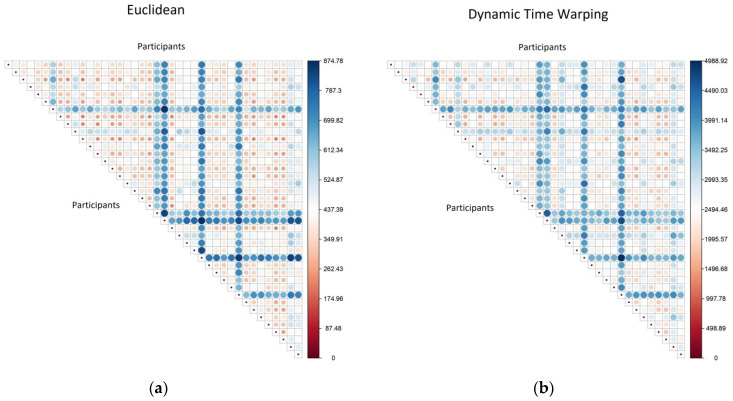
A comparison of the distance matrices calculated using two different distance metrics, with participant pairs: (**a**) calculated using a Euclidean metric; (**b**) calculated using a dynamic time warping (DTW) metric.

**Figure 3 sensors-21-08220-f003:**
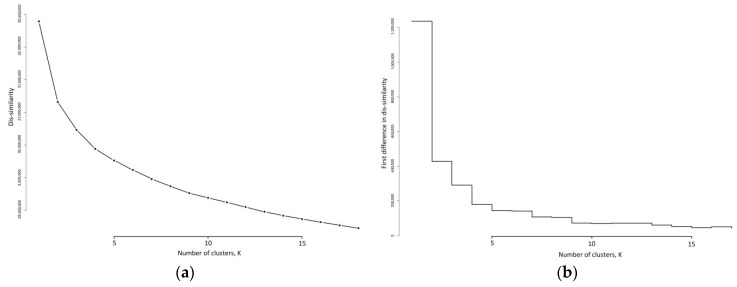
Dis-similarity metrics as the number of clusters, k, increases: (**a**) as dis-similarity; (**b**) as the first difference in dis-similarity.

**Figure 4 sensors-21-08220-f004:**
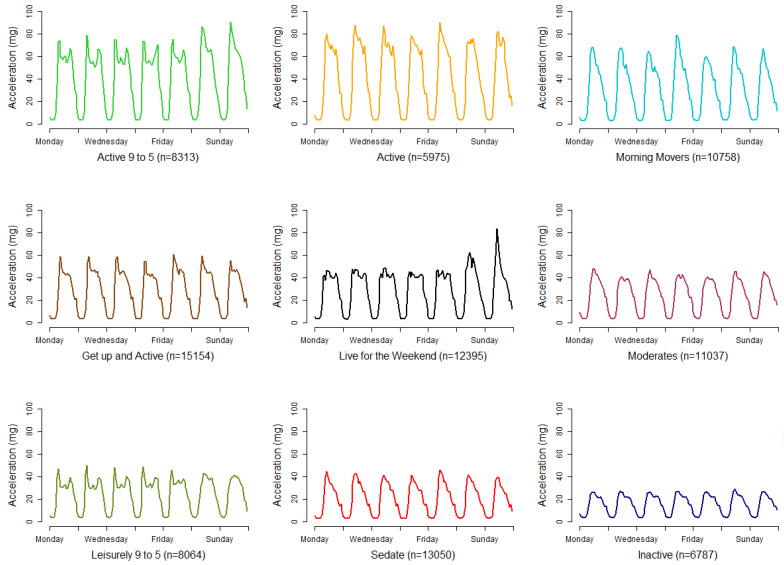
Average profiles for activity patterns associated with each cluster from a nine-cluster solution. A label is associated with each profile and is shown below each chart along with n, the number of participants in each cluster.

**Table 1 sensors-21-08220-t001:** Comparison of the composition of the full UK Biobank sample, those contributing accelerometer data, and those available for clustering.

Characteristic	Clustering Sample	Accelerometer Sample	UK Biobank
Female	56.5%	56.2%	54.4%
Younger, 40 to 54	39.6%	39.1%	38.8%
White/British	96.6%	96.4%	94.1%
In paid employment or self-employed	61.8%	62.3%	57.4%
Non-Car/motor vehicle commute	39.9%	40.0%	63.0%
College/University	42.9%	43.1%	32.1%
School qualifications	37.5%	37.4%	37.3%
Income of more than £31,000	54.8%	55.1%	43.9%
Least 20% deprived	44.5%	44.2%	39.9%
Healthy BMI	38.7%	38.6%	32.3%
Excellent/good health	81.5%	81.3%	73.8%
No Long standing illness	70.1%	70.1%	65.6%
Very/Fairly easy to get up	82.2%	82.1%	81.0%
Definitely/more a ‘morning’ person	56.5%	56.2%	55.3%
Spring/summer accelerometer wear	48.8%	49.2%	NA
N	91,533	103,332	500,028

**Table 2 sensors-21-08220-t002:** Composition of each activity pattern, with a heatmap showing higher values in red and lower values in blue.

Characteristic	Active 9 to 5	Active	Morning Movers	Get Up and Active	Live for the Weekend	Moderates	Leisurely 9 to 5	Sedate	Inactive	Clustering Sample
Female	61.1%	60.2%	59.5%	61.2%	56.0%	58.2%	49.5%	52.8%	45.3%	56.5%
Younger, 40 to 54	67.8%	46.4%	32.6%	31.6%	49.6%	35.8%	66.1%	18.0%	26.2%	39.6%
White/British	95.4%	97.2%	97.5%	97.2%	96.3%	95.9%	94.1%	98.0%	96.5%	96.6%
In paid employment or self-employed	81.6%	65.3%	55.8%	56.3%	71.1%	59.2%	85.8%	44.7%	47.8%	61.8%
Non-car/motor vehicle commute	46.6%	40.6%	36.5%	41.3%	39.2%	39.3%	41.9%	34.4%	36.3%	39.9%
Attended college/university	46.9%	41.9%	39.2%	44.3%	46.1%	44.8%	46.5%	37.4%	39.6%	42.9%
School qualifications	39.3%	40.6%	38.8%	36.2%	37.6%	36.7%	38.0%	36.8%	35.2%	37.5%
Income of more than GBP 31,000/year	66.0%	56.3%	52.3%	52.9%	62.4%	51.7%	67.2%	44.7%	43.6%	54.8%
Lives in east 20% deprived neighbourhood	43.4%	46.5%	47.7%	47.3%	45.2%	41.6%	39.1%	46.8%	38.1%	44.5%
Healthy BMI	55.8%	54.1%	43.5%	42.5%	41.1%	33.6%	33.7%	27.2%	20.1%	38.7%
Excellent/good health	89.8%	88.7%	86.5%	84.9%	85.0%	77.4%	80.4%	76.6%	60.9%	81.5%
No long standing illness	81.5%	77.7%	74.6%	71.4%	74.9%	66.3%	72.7%	62.5%	48.1%	70.1%
Very/fairly easy to get up	84.9%	81.1%	85.4%	85.1%	84.6%	72.5%	82.7%	84.4%	75.1%	82.2%
Definitely/more a morning person	65.7%	52.5%	60.2%	60.5%	61.7%	38.8%	63.1%	56.6%	44.6%	56.5%
Spring/summer accelerometer wear	54.4%	52.3%	49.5%	48.4%	51.0%	47.0%	49.7%	44.4%	44.9%	48.8%
N (%)	8313 (9.1%)	5975 (6.5%)	10,758 (11.8%)	15,154 (16.6%)	12,395 (13.5%)	11,037 (12.1%)	8064 (8.8%)	13,050 (14.3%)	6787 (7.4%)	91,533 (100%)

**Table 3 sensors-21-08220-t003:** Composition of weight status summarised by percentage in each activity profile pattern, and in comparison to the total cohort.

	Active 9 to 5	Active	Morning Movers	Get Up and Active	Live for the Weekend	Moderates	Leisurely 9 to 5	Sedate	Inactive
Healthy	13.1% (+4.0%)	9.1% (+2.6%)	13.2% (+1.5%)	18.2% (+1.6%)	14.4% (+0.8%)	10.5% (−1.6%)	7.7% (−1.1%)	10.0% (−4.2%)	3.8% (−3.6%)
Overweight	7.6% (−1.5%)	5.8% (−0.7%)	12.0% (+0.3%)	16.8% (+0.3%)	13.8% (+0.2%)	12.6% (+0.6%)	8.8% (0.0%)	15.6% (+1.4%)	7.0% (−0.4%)
Obese	4.0% (−5.0%)	2.8% (−3.7%)	8.4% (−3.4%)	12.9% (−3.7%)	11.4% (−2.1%)	14.1% (+2.1%)	11.1% (+2.3%)	19.9% (+5.7%)	15.3% (+7.9%)
All	9.1%	6.5%	11.8%	16.6%	13.5%	12.1%	8.8%	14.3%	7.4%

**Table 4 sensors-21-08220-t004:** Composition of each activity pattern by COVID-19 outcomes, with a heatmap showing higher values in red and lower values in blue.

COVID-19 Outcomes	Active 9 to 5	Active	Morning Movers	Get Up and Active	Live for the Weekend	Moderates	Leisurely 9 to 5	Sedate	Inactive	Clustering Sample	Wearable Sample	UK Biobank
Alive on 23 March 2021	8215	5891	10,534	14,801	12,153	10,696	7897	12,455	6205	88,847	100,292	465,472
% Participants alive	98.8%	98.6%	97.9%	97.7%	98.0%	96.9%	97.9%	95.4%	91.4%	97.1%	97.1%	93.1%
% Participants alive and tested	16.1%	14.4%	16.1%	16.6%	17.3%	17.9%	17.1%	18.0%	20.8%	17.1%	17.2%	18.6%
% Participants alive and tested positive	3.7%	2.9%	2.8%	2.4%	3.1%	2.8%	4.0%	2.3%	3.0%	2.9%	3.0%	3.7%
Positive rate	23.20%	20.4%	17.2%	14.6%	17.8%	15.7%	23.19%	12.8%	14.4%	17.0%	17.2%	20.1%

## Data Availability

The data used in this study were obtained from the UK Biobank (https://www.ukbiobank.ac.uk/ accessed: 7 December 2021) under application and project ID 30846. Individuals wishing to replicate this research should contact the UK Biobank and request these data. The code for analysis can be obtained from the authors.

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
