# Peer review of "Clustering Accelerometer Activity Patterns from the UK Biobank Cohort"

_sensors, 2021, doi:10.3390/s21248220_

Round 1

Reviewer 1 Report

In this study, the authors used an unsupervised machine learning approach, K-Medoids, to explore activity patterns based on accelerometer data from an extensive one-week accelerometer data. The results of 9 activity patterns are interesting and may contribute to a better understanding of other health and morbidity outcomes.

1)The introduction can be improved. It is reasonable that physical activity levels are associated with severity levels of Covid-19, but why physical activity levels are associated with positive rate?

2) Generally, in the field of machine learning, k-means is a clustering method that is unsupervised learning. Classification belongs to supervised learning, which means that the data are labeled. The author used K_Medoids function from the R package ClusterR to cluster the acceleration sequences into 9 clusters and called K_Medoids as classification method. In my opinion, it is unsuitable. Please rephrase the title and corresponding sentences.

3) The process of raw accelerometer data is unclear. The authors ref [51] which provided acceleration values (mg) for typical daily activities and the acceleration was computed by ENMO. So, did you use ENMO or remove the gravity component in the current study?

4) The authors should evaluate the clustering performance of k-medoids on the current dataset.

5) Table 2 is not clear, how can you label (stigmatize) groups of people based on the activity pattern this seem to much of a short cut, similar the way the relation is made with BMI.

Specific comments:

1) line 24, what is "90k" mean?

2) from line 91-94, the sentence is too long. Please rephrase it.

3) line 102, what is "500k" mean?

4) line 159, please explain why 100mg was selected as a threshold.

5) from line 257-259. The sentence is unclear. Are the profiles in Figure 1 excluded from the study? In Figure 1 (a), no sleeping time can be recognized. This is invalid data. 

6) figure 2 is unreadable

Reviewer 2 Report

In this manuscript, the authors have identified nine patterns of physical activity from a very large cohort sourced from the UK Biobank who wore wrist worn accelerometer for a week. The manuscript is well written, technical methods are sound and I believe the readers of Sensors will find the topic of this work interesting. I would like to thank the authors for the opportunity to review their work and hope that my comments help improve the manuscript for the readers.

Major concerns

I found the method for identifying the classifications well written and justified. However, my main concern is with the second part of the manuscript. It is unclear how the classifications are useful above and beyond just describing the level and timing of activity as separate variables (and possible with interaction terms depending on the analysis). That is, what is the benefit of spending the time and computational cost to quantify clusters over and above the data we already have available? I strongly recommend that the authors either: i) reword the arguments about the utility or meaningfulness of the analysis as this has not been shown robustly in this manuscript; or ii) add additional analysis to show usefulness above and beyond what is already available.

Minor concerns

Although I appreciate the active language used in some parts of the introduction makes for more engaging reading, I recommend rephrasing or removing hyperbole such as “Physical activity could be described as a super power…” or more emotive statements such as “At a time of climate crisis and such clear benefits associated with physical activity it is difficult to understand why so many remain insufficiently active.” This phrasing does not help in objective communication of this scientific work.

There are also sentences with multiple statements and it is not clear which the citations refer to, such as “In the developed world, where manual labour employment is less prevalent, physical inactivity rates are higher, …”. Also, there is no citation for line 43-44 “This comes not only …” I assume this is also from the Public Health England citation as well.

Typos: line 54 “physical”

Line 353 “particularly”

Line 452 – do you mean sex or gender?

Line 546 – do you mean higher instead of increased?

Potentially of interest to readers (but not a concern per se)

I suggest adding details of how the classification described in this manuscript can be applied to future datasets, i.e. cutoffs or decision trees. (my apologies if I have missed this)?

Also the classification with respect to timing appears to rely on weekday, Saturday & Sunday activity. As such, do the authors believe that accurate classification with only three days of activity monitoring (weekend + one weekday) is feasible? This would be useful for those planning future studies for others in the field wanting to use your classifications.

Round 2

Reviewer 1 Report

The authors have addressed the comments adequately.

Reviewer 2 Report

The authors have addressed the concerns I raised in my first review. I commend the authors for their work and contribution to the field.